# Running-Induced Fatigue Changes the Structure of Motor Variability in Novice Runners

**DOI:** 10.3390/biology11060942

**Published:** 2022-06-20

**Authors:** Felix Möhler, Cagla Fadillioglu, Lucia Scheffler, Hermann Müller, Thorsten Stein

**Affiliations:** 1BioMotion Center, Institute of Sports and Sports Science (IfSS), Karlsruhe Institute of Technology, 76131 Karlsruhe, Germany; cagla.fadillioglu@kit.edu (C.F.); lucia.scheffler@gmx.de (L.S.); thorsten.stein@kit.edu (T.S.); 2Training Science, Department of Sports Science, Justus-Liebig-Universität Giessen, 35394 Giessen, Germany; hermann.mueller@sport.uni-giessen.de

**Keywords:** center of mass, motor control, biomechanics, uncontrolled manifold (UCM)

## Abstract

**Simple Summary:**

Endurance sports, and especially running, are very popular. During running, fatigue inevitably occurs, and especially in novices. Surprisingly, the effects of fatigue have been studied less extensively in novice runners compared with experienced runners. Regardless of the level of expertise, it has been shown that, by analyzing motor variability, valuable insights can be gained regarding the control of important variables. Motor variability in running is understood as step-to-step deviations. One example of an important variable is the body’s center of mass, as it provides a simplified representation of the overall movement. Thus, an analysis of the motor variability of the body’s center of mass might lead to valuable insights. Therefore, the purpose of this study was to investigate the effects of fatigue on the motor variability of the body’s center of mass in novice runners. It was found that, with fatigue, the motor variability increased, and the control of the body’s center of mass decreased. Moreover, there was a correlation between the decrease in control and the degree of fatigue. Further studies should investigate which training methods can mitigate this effect.

**Abstract:**

Understanding the effects of fatigue is a central issue in the context of endurance sports. Given the popularity of running, there are numerous novices among runners. Therefore, understanding the effects of fatigue in novice runners is an important issue. Various studies have drawn conclusions about the control of certain variables by analyzing motor variability. One variable that plays a crucial role during running is the center of mass (CoM), as it reflects the movement of the whole body in a simplified way. Therefore, the aim of this study was to analyze the effects of fatigue on the motor variability structure that stabilizes the CoM trajectory in novice runners. To do so, the uncontrolled manifold approach was applied to a 3D whole-body model using the CoM as the result variable. It was found that motor variability increased with fatigue (UCM_ꓕ_). However, the UCM_Ratio_ did not change. This indicates that the control of the CoM decreased, whereas the stability was not affected. The decreases in control were correlated with the degree of exhaustion, as indicated by the Borg scale (during breaking and flight phase). It can be summarized that running-induced fatigue increases the step-to-step variability in novice runners and affects the control of their CoM.

## 1. Introduction

The popularity of running has been expanding worldwide for many years. The number of participants in official running competitions increased from less than 2 million in 2001 to more than 7.9 million by 2018 [1]. The goal of runners, whether they are at the novice, recreational or expert level, is to improve their fatigue resistance [2]. However, the mechanisms and effects of fatigue are not yet fully understood [3]. In addition, research concerning novice runners is sparse, despite its relevance to the high number of people who begin to run without prior experience. Furthermore, novice runners were shown to be prone to injury [4]. Even though expert runners have higher exposure and, thus, greater risk for injuries due to the wear of structures, novice runners probably do not have an adequate running style yet and may therefore load their joints, as well as their muscles, in an unbalanced manner. 

The interest in the influence of fatigue on running biomechanics has inspired numerous studies [5,6,7,8,9,10,11,12]. However, the study results are partly conflicting and further investigations are needed for a better understanding [3]. For example, Rabita et al. [13] showed that, under fatigue at a constant speed, the step frequency increases, whereas the vertical displacement of the center of mass (CoM) decreases. In a similar study [14], however, no changes in the step frequency or vertical displacement of the CoM were detected. 

In running-related studies, spatiotemporal parameters and joint angles [3] have frequently been analyzed to reveal changes in the kinematics, and motor variability has been analyzed less often [15]. Besides these kinematic measures, the position of the CoM is one of the key parameters. Although the CoM represents a theoretical construct, it is used to describe human motion because it reflects the movement of the whole body [16]. It is thus considered to be an important parameter for locomotion [17,18]. It has been suggested that the CoM is generally controlled by adapting the leg parameters during running [19].

Recently, it has been shown that running-induced fatigue does not affect the spatiotemporal parameters or their variability in novice runners, although the position of the CoM is affected [20]. This is in contrast to findings in expert runners, which have shown that a high-intensity run with a similar time to exhaustion affects the spatiotemporal parameters and their variabilities [21]. The changes in the CoM position that occur with the fatigue seen in novice runners (a lower position around the heel strike and an increased range of motion (RoM) in the medio-lateral direction) might be explained by a loss of control of the CoM. In expert runners, it was found that, with increasing fatigue, the running speed was diminished rather than the height of the CoM [7]. Strohrmann et al. [12] found longer ground-contact times and greater vertical displacement in novice runners, whereas experts showed less change in running technique during a 45 min exhausting run on the treadmill, although the variabilities in the joints were not included in the study. Additionally, Mo and Chow [10] demonstrated that there is less joint and segmental kinematic variability in experienced runners under fatigue than in novice runners, but they did not analyze how the variability relates to the CoM position. To the best of our knowledge, the motor variability in multijoint movements in association with the CoM trajectory in novice runners has not yet been analyzed. 

The musculoskeletal system deals with phenomena such as noise in signal processing, nonlinearities, an enormous redundancy and others. These features result in motor variability. This led to the famous observation of Bernstein, who stated that one specific movement cannot deliberately be repeated in the exact same way [22]. However, specific variables, which are crucial for the movement outcome, are constant over repetitions. In Bernstein’s study of blacksmiths, the positions of the shoulder, elbow and wrist changed over repeated movement executions, but the position of the hammer tip remained constant. Apparently, variability is permitted as long as it does not impair the movement goal. However, variability at points where it might impair the movement goal seems to be restricted, which has been interpreted as an indication of control [23]. 

The control of specific variables has been quantified using the uncontrolled manifold (UCM) hypothesis, developed by Scholz and Schöner [24]. By analyzing variability over several movement executions or repetitions, a control hypothesis about a so-called result variable (RV) is tested [24]. The RV, also referred to as the performance variable (e.g., in [25]), is a variable of which the value is crucial for the movement outcome. One possible RV in the case of locomotion could be the CoM [26,27,28]. This RV is thus hypothesized to be stabilized by several elemental variables (EVs) (for example, joint angles) [26,27,28,29,30]. 

The RV can be stabilized through covariance among the EVs, which compensates for individual deviations. This means that a deviation in one EV can be compensated for by a corresponding change in another EV. In this way, there is variability across the movement executions that does not affect the RV; thus, with respect to the RV, equivalent movement solutions exist. This portion of motor variability that does not affect the RV is also referred to as UCM_‖_. It is considered beneficial because it constitutes a source of flexible movement executions by providing a multitude of equivalent movement solutions. If changes in an EV are not compensated for by another EV, changes in the RV occur. The motor variability that leads to changes in the RV is referred to as UCM_ꓕ_. This portion of variability is potentially “unwanted”, as it affects the RV. A low level of UCM_ꓕ_ is thus interpreted as an indication of a high degree of control. If the UCM_‖_ is greater than the UCM_ꓕ_, the chosen RV is assumed to be stabilized by the central nervous system, and the control hypothesis about the RV is accepted. The ratio of the UCM_‖_ to the UCM_ꓕ_ (UCM_Ratio_) is used to quantify the degree of stabilization [23,25].

The UCM has yielded valuable results in a number of studies. Studies that analyzed the total force applied by the fingers as the RV showed that the values of the UCM_‖_ and UCM_ꓕ_ increased under fatigue, and therefore increased the variability in the EVs [31,32]. In another study that analyzed the gender-specific effects of fatigue on lower-limb stability during a pointing task, higher UCM_‖_ and UCM_ꓕ_ were reported for women in comparison with men in a fatigued state [33]. The UCM approach has been used several times for locomotion studies (e.g., Qu et al. reported a significantly lower value of the UCM_Ratio_ in the frontal plane during walking in a fatigued state than in a recovered state [28]), which means that the CoM is less stabilized in the fatigued state than in the recovered state. Möhler et al. showed that, even under fatigue, the CoM trajectory of expert runners was stabilized over the entire gait cycle [26]. While the UCM_Ratio_ and UCM_‖_ were unaffected by fatigue, the UCM_ꓕ_ increased in the flight phase. In addition, Möhler et al. [11] showed that, at higher speeds, the motor variability during running is higher in novices than in experts.

The UCM is a promising approach to gain further insight into the effects of fatigue on the variability structure that stabilizes the CoM in novices during running. Therefore, the aim of the present study was to analyze the influence of fatigue on the stride-to-stride motor variability structure that stabilizes the CoM trajectory in novice runners. It was hypothesized that novice runners would lack strategies to control their CoM in a fatigued state and would not stabilize their CoM trajectories, which would be indicated by a decreased UCM_Ratio_ and increased UCM_ꓕ_, respectively. The changes in these UCM parameters would correlate with the level of fatigue. 

## 2. Materials and Methods

This study reanalyzes the data of a previously published study [20]. The details of the participants, experimental protocol and data collection are repeated in the following subsections. 

### 2.1. Participants

A total of 14 healthy young novice runners (age: 27.4 ± 4.3 years; stature: 1.82 ± 0.06 m; body mass: 77.5 ± 10.3 kg; running activity per week: 14 ± 18 min; other sports activity per week: 110 ± 71 min) participated in the study. Exclusion criteria were regular running exercise more than once a month, any injury or pain in the lower limbs within the last six months prior to data collection and a BMI higher than 25 kg/m^2^. All participants provided written informed consent to voluntarily participate in this study. The study was approved by the ethics committee of the Karlsruhe Institute of Technology. 

### 2.2. Experimental Protocol

The measurements were conducted in the Biomechanics Laboratory of the BioMotion Center at the Institute of Sports and Sports Science of the Karlsruhe Institute of Technology. The participants ran on a treadmill (h/p/cosmos Saturn, Nussdorf-Traunstein, Germany) with a slope of 1% [34]. The participants were first familiarized with the treadmill by walking at a speed of 5 km/h for 6 min [35] and running at a speed of 8 km/h for 6 min [36], followed by 10 s of running at a speed of 13 km/h. After the treadmill familiarization, the subjects had two minutes to recover before running at a fixed speed of 13 km/h until subjective exhaustion. The speed of 13 km/h was chosen by experience as a speed that would challenge novice runners sufficiently to lead to exhaustion after several minutes. Participants were instructed to look straight ahead and to not perform undesired movements. To prevent falls, all participants wore a safety harness during the experiment. The treadmill was stopped immediately when the participants indicated exhaustion. After having indicated exhaustion and after the treadmill stopped, the participants were asked to rate their fatigue on the Borg scale [37]. 

### 2.3. Data Collection and Processing

First, 22 anthropometric measures were manually taken from each participant, and 42 reflective markers were attached to participants’ skin, in accordance with the ALASKA modeling system (Advanced Lagrangian Solver in Kinetic Analysis, INSYS GmbH, Chemnitz, Germany; [38]). During the treadmill protocol, 16 Vicon cameras (Vicon Motion Systems; Oxford Metrics Group, Oxford, UK) recorded the subjects’ kinematics, with a recording frequency of 200 Hz.

Data recording started 10 s after the treadmill reached the speed of 13 km/h and lasted until the treadmill stopped. The marker data were preprocessed using Vicon Nexus V2.11.0 software and filtered with a second-order low-pass Butterworth filter, with a cutoff frequency of 15 Hz, using MATLAB R2020b (The MathWorks, Natick, MA, USA). The anthropometric measurements (22 measured manually, 43 determined from the reflective markers, according to the requirements of the ALASKA modeling system) and the marker trajectories were used to calculate joint angles using inverse kinematics with the full-body Dynamicus model (ALASKA, INSYS GmbH, Chemnitz, Germany; [38]).

From the recorded data, the first 35 gait cycles represented the PRE condition (rested state), and the last 35 gait cycles represented the POST condition (fatigued state). Gait cycles were detected based on the sign change of the heel or forefoot marker and the vertical acceleration of the toe marker at initial contact and toe off, respectively [39]. For the subsequent UCM analysis, each gait cycle was time-normalized to 101 data points using cubic spline interpolation. 

### 2.4. Uncontrolled Manifold Approach

Variability analyses may be sensitive to the number of gait cycles included [40,41]. Considering the application of UCM purely in terms of the mathematical aspect, more movement repetitions would provide more reliable results [40]. However, in the present case, the fatigue effects might interfere when including too many cycles. Therefore, *N*_cycle_ = 35 was chosen as the number of gait cycles to be included in the present analysis. 

The necessary steps for the UCM approach were first explained by Scholz and Schöner [24]. The calculations used in this study, with specific consideration to the 3D CoM, are outlined in the following section.

Due to the importance of the CoM in running [16], and in line with other studies [11,26,27,28,42,43,44,45], the CoM trajectory was chosen as the RV, and the joint angles were chosen as EVs. 

To calculate the CoM based on the EVs, a subject-specific anthropometric 3D model consisting of 17 segments and 50 degrees of freedom (47 segmental angles and 3 hip rotations [26]) was used. The whole-body CoM ($r_{\mathrm{CoM}}$) was calculated as a weighted sum of the body segments (e.g., [46]), as in Equation (1):(1)$$r_{\mathrm{CoM}}=\frac{1}{\sum_{i=1}^{N}V_{i}}\times \sum_{i=1}^{N}r_{i}V_{i},$$
where *N* is the number of segments; $V_{i}$ is the volume of the *i*th segment; and $r_{i}$ is the center-of-gravity vector of the *i*th segment.

This model enabled changes in joint angles (*Θ*, EV) to be linked to changes in the CoM trajectory ($r_{\mathrm{CoM}}$, RV), where the RV was expressed as a function of the EVs, as in Equation (2). Because the EVs in a UCM model must have the same unit, only joint angles were used, instead of different quantities (e.g., joint angles combined with kinematic variables; see also [47]): (2)$$\mathrm{RV}=r_{\mathrm{CoM}}=f(\mathrm{EV})=f(\Theta ).$$

The space in which changes in the EVs do not cause changes in the RV corresponds to the null space of the linearized Jacobian. According to the UCM approach [24], it is calculated as in Equation (3): (3)$$0=J_{e_{i}}={\frac{\partial f\left(\Theta \right)}{\partial \Theta }|}_{\Theta_{0}}$$
with *i* = 1 … *n*-*d*, where *n* is the number of dimensions of EVs, and *d* is the number of dimensions of the RV (here: *n* = 50 and *d* = 3, respectively); $\Theta_{0}$ are the mean values of the EVs over the 35 gait cycles, and $e_{i}$ are the vectors defining the null space. 

In the next step, deviations from $\Theta_{0}$ were separated into those parallel to the UCM (stabilizing the RV (*σ*_k,‖_), Equation (4)): (4)$$\sigma_{k,\|}=\sum_{i=1}^{n-d}\left[\left(e_{i}^{T}\left(\Theta_{k}-\Theta_{0}\right)\right)e_{i}\right]$$
and those orthogonal to the UCM (changing the RV (*σ*_k,__ꓕ_), Equation (5)): (5)$$\sigma_{k,ꓕ}\;=\left(\Theta_{k}-\Theta_{0}\right)-\sigma_{k,\|},$$
where k = 1, …, *N*_cycle_ and *N*_cycle_ is the number of included gait cycles (here: *N*_cycle_ = 35). These calculations were performed for each time point of the normalized gait cycle, which led to a total of 101 UCM calculations. Subsequently, the variabilities parallel and orthogonal to the UCM were calculated as the variance over the 35 gait cycles, as in Equation (6) and Equation (7), respectively:(6)$${\mathrm{UCM}}_{\|}=\sqrt{\frac{1}{\left(n-d\right)*N_{\mathrm{cycle}}}\sum_{k=1}^{N_{\mathrm{cycle}}}o_{k,\|}^{2}}$$
and
(7)$${\mathrm{UCM}}_{ꓕ}\;=\sqrt{\frac{1}{d*N_{\mathrm{cycle}}}\sum_{k=1}^{N_{\mathrm{cycle}}}o_{k,⊥}^{2}}.$$

The ratio between these two quantities was calculated: (8)$${\mathrm{UCM}}_{\mathrm{Ratio}}=\frac{2*\;UCM_{\|}^{2}}{UCM_{\|}^{2}+UCM_{⊥}^{2}}-1.$$

The UCM_Ratio_ quantifies the degree of stabilization of the RV (in this study, the CoM). It has a theoretical range of [−1, 1], where a positive value indicates a stable state, and vice versa [42,43]. 

### 2.5. Statistics 

Statistical analyses were performed using MATLAB R2020b (The MathWorks, Natick, MA, USA). First, all data were checked for normal distribution using the Shapiro–Wilk test [48]. The time courses were tested for differences between the PRE and POST for each dependent parameter (UCM_‖_, UCM_ꓕ_ and UCM_Ratio_) using a dependent *t*-test with statistical parametric mapping (SPM) from the spm1d toolbox [49]. The significance level was set at *p* = 0.05. Afterwards, the entire gait cycle was divided into two flight phases (FPs), and the left and right stance phases, according to the detected gait events [39]; both stance phases were divided into a breaking phase (BP) and a propulsion phase (PP) [50]. Right and left flight phases, as well as right and left breaking and propulsion phases, were then averaged to calculate one value for each phase in the following abbreviations: FP, BP and PP. Afterwards, the changes from PRE to POST (Δ_Pre-Post_) in the mean values of each UCM parameter were calculated by subtracting the mean value of POST from the mean value of PRE for the three phases of the gait cycles: FP, BP and PP. Spearman correlations between the Δ_Pre-Post_ and the Borg-scale rating were separately calculated for each UCM parameter and each phase of the gait cycle individually. These calculations were performed as a cross-check to assess whether the UCM parameters can reflect the effects of fatigue captured by the Borg-scale ratings. 

## 3. Results

The participants maintained the speed of 13 km/h for 6.18 ± 2.45 min. Their exhaustion was confirmed by a Borg-scale rating of 18.7 ± 1.0 (scale ranging from 6 to 20), which corresponds to “very very hard” in terms of the difficulty level.

The UCM analysis showed that both the UCM_ꓕ_ and UCM_‖_ were higher in the POST state (see Figure 1) throughout the gait cycle. However, the differences concerning the UCM_‖_ did not reach the level of statistical significance, whereas the differences for the UCM_ꓕ_ did (see Figure 2). The UCM_Ratio_ did not show any statistically significant changes throughout the gait cycle between the PRE and POST. In both states, it was greater than zero throughout the entire gait cycle.

To investigate the relationship between the changes in the UCM parameters from the PRE to the POST (Δ_Pre-Post_) and the level of exhaustion, Spearman correlations (ρ) were calculated. Table 1 shows the UCM_ꓕ_ for the BP and FP correlated with the Borg scale, with *p* values of 0.055 and 0.032, respectively. 

## 4. Discussion

The aim of our study was to analyze the influence of fatigue on the stride-to-stride motor variability structure that stabilizes the CoM trajectory in novice runners. Therefore, a UCM analysis was performed in which the CoM was chosen as the RV. The results revealed significant increases in the UCM_ꓕ_, but no significant effects on the UCM_‖_ or UCM_Ratio_. The changes in the UCM_ꓕ_ can be interpreted as decreased control over the CoM [24]. However, this decrease did not affect the stability of the CoM, as revealed by the unaffected UCM_Ratio_. The fact that the UCM_Ratio_ was constantly above zero shows that the CoM is controlled throughout the whole gait cycle in both the PRE and POST states. The variability in terms of the UCM_ꓕ_ was increased in the POST state, which is in line with previous studies [3,26]. Furthermore, the increases in the UCM_ꓕ_ correlated with the Borg-scale ratings in the BP and FP (*p* = 0.055 and *p* = 0.032, respectively), which indicated the plausibility of using UCM analysis for analyzing the control of the CoM trajectory under fatigue.

In our previous study [20], we showed that novice runners had a lower CoM position around the heel strike and increased RoM of the CoM in the medio-lateral direction in the POST state. Combined with the results from this study, it may be concluded that the decreased control of the CoM might be the reason for the changes in the CoM trajectory. Furthermore, in the present study, increases in the UCM_ꓕ_ after the right and left heel strikes were observed (Figure 1).

Variability is an important and ubiquitous feature of human movement. It has been hypothesized that decreased variability could indicate a tendency for overuse injury [51]. When studying variability, the level of analysis of the motor variability is crucial. For instance, a minimum variability in the level of joint angles could be important to avoid overuse injuries, whereas keeping the step frequency constant may be important for an efficient running style [52]. However, this does not necessarily constitute a contradiction. As assessed within the UCM approach, the covariation of individual parameters can be used to provide flexible movement variants while keeping an RV constant [23]. This can be interpreted as a compensatory mechanism for stabilizing the RV. In the present study, and in our previous ones, we have shown that UCM with a 3D whole-body model is applicable to running movement for stride-to-stride analyses, and that this method provides meaningful results [11,26]. The fact that the changes in the UCM_ꓕ_ observed in this study correlated with the Borg-scale rating of the participants supports the plausibility of applying the UCM approach to study the effects of fatigue on the control of the CoM trajectory. To deepen our understanding of the compensatory mechanisms, one possibility could be to relate the UCM parameters to the movement patterns and their changes, which would help us to find out which EV has the largest effect on the changes in the RV. Mathematically, this information is contained in the Jacobian matrix, as it relates the rate of the changes in all the EVs to changes in the RV. However, there is a separate Jacobian for each time step, and a separate entry for each dimension of the RV. In this entry, there is the contribution of all the EVs. Due to this complexity, the extraction of the desired information is not straightforward. 

The fact that the UCM_Ratio_ is above zero is interpreted as an indicator for the control of the RV [42,43], which, in our case, is the CoM. Although it has been suggested that the CoM is a controlled variable, and especially in the postural studies [53], it is hard to verify this experimentally. Scholz et al. [54] showed that, during balance recovery, participants tended to re-establish the position of the CoM rather than those of the joint configurations. Consequently, they suggested that the CoM is the key variable controlled by the central nervous system. In the present study, overall, both the UCM_ꓕ_ and UCM_‖_ were higher in the fatigued state, indicating more variable joint angles.

Because UCM analysis examines variances from trial to trial, the number of repetitions (e.g., gait cycles) taken into consideration plays a crucial role in obtaining reliable results [40]. If fatigue effects are studied by using the UCM approach, then subjects cannot sustain the performance indefinitely. However, there are currently only a few studies that address this issue. The number of repetitions needed varies widely depending on the movement task, method and UCM parameter [40,41,55,56]. The large variations can be explained by the inherent noise in the original signals. The kinematics of a multi-joint movement, such as running, are generally associated with higher noise than the kinematics of a single-joint movement, such as elbow flexion. Therefore, even though 15–20 cycles have been recommended for analyzing gait data, we chose to include 35 cycles in our analysis [41]. 

There are some limitations that should be considered. (1) When studying variability, performing a study on a treadmill may be a drawback, since a certain amount of variability is abolished due to the fixed running speed. On the one hand, the runners were not able to adapt the running speed, which is contrary to real-world scenarios. On the other hand, it has been shown that running style depends on running speed [16]. Moreover, if the speed is variable, then it is difficult to distinguish between the motor variability and the effects due to the changes in the speed. [57]. When studying the effects of fatigue, it is thus preferable to keep the running speed constant, even though reducing the running speed may be a possible strategy to maintain the CoM trajectory [7]. (2) Even though the Borg scale is a standardized method, a purely subjective measure may be considered as a limitation. On the other hand, a subjective measure might be better able to capture the state of fatigue resented by the participants, as fatigue has many facets. (3) In this study, the CoM was chosen as an RV. However, it was not necessarily the single right one for such studies. A multitude of parameters regarding efficiency, injury prevention and performance could also be of interest, as these are important during running and should therefore be stabilized or held constant. (4) When calculating the SPM over the whole gait cycle, an error might occur because the ratio between the stance- and swing-phase durations might change with fatigue [21]. However, in our previous study, we found no changes in the spatiotemporal parameters. Therefore, any bias due to temporal shifts in the normalization process can be ignored.

## 5. Conclusions

Thousands of people start running every year. Although the motivations are numerous, one that is common to many novice runners is to improve their fitness and, therefore, to push themselves to their limits. The occurrence of fatigue is thus inevitable. Understanding the effects of fatigue on movement patterns and its variability is crucial. By using the UCM approach, the effects of fatigue on the variability structure were analyzed in the present study by looking at the CoM trajectory as a crucial parameter in locomotion. In a fatigued state, the control of the CoM was decreased, but its stability was unaffected. The decrease in control was correlated with the ratings of perceived exertion. In contrast to expert runners, the structure of the motor variability in novices is thus affected by fatigue. 

Because the UCM was able to capture the effects due to fatigue on the CoM trajectory, it may be used to evaluate the outcome of running-technique training, as the aim of technique training is ultimately to improve the running style, and the running style can be operationalized by the CoM trajectory [16]. 

## Figures and Tables

**Figure 1 biology-11-00942-f001:**
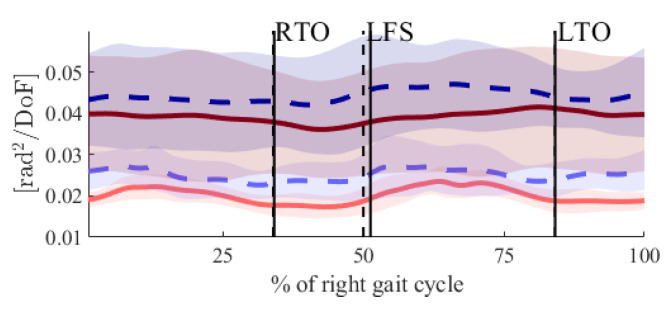
Time courses for UCM_ꓕ_ (light-red and light-blue lines) and UCM_‖_ (dark-red and dark-blue lines), PRE (solid lines) and POST (dashed lines) fatigue. The perpendicular black lines show the gait events right toe off (RTO), left foot strike (LFS) and left toe off (LTO). Solid lines are the gait events PRE fatigue, and dashed lines are the gait events POST fatigue.

**Figure 2 biology-11-00942-f002:**
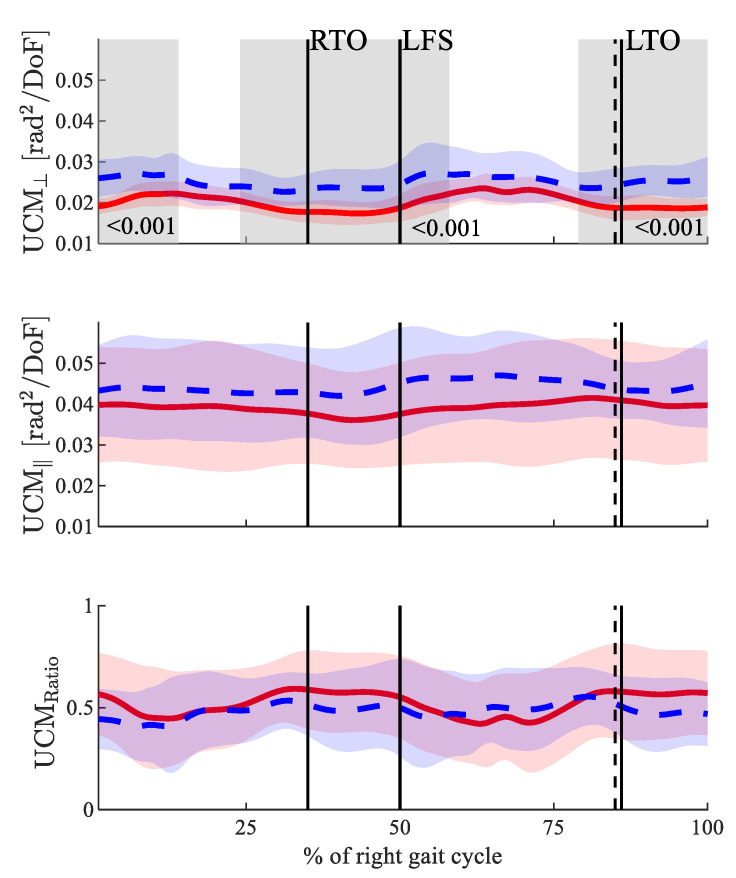
Results for the three dependent UCM parameters. From top to bottom: UCM_ꓕ_, UCM_‖_ and UCM_Ratio_. The solid red line shows the results PRE fatigue, and the dashed blue line shows the results POST fatigue. The perpendicular black lines show the gait events right toe off (RTO), left foot strike (LFS) and left toe off (LTO). Solid lines are the gait events PRE fatigue, and dashed lines are the gait events POST fatigue. Significant differences are highlighted in grey.

**Table 1 biology-11-00942-t001:** Correlations (ρ ) and *p*-values (*p*), as calculated by Spearman correlations, between the changes in UCM parameters and the Borg scale. ΔPre-Post values are given as mean ± standard deviation over the participants. BP, PP and FP signify the breaking, propulsion and flight phases, respectively.

Dependent Parameter	Phase	ΔPre-Post	ρ	*p*
UCM_ꓕ_	BP	0.005 ± 0.003	0.523	0.055
	PP	0.004 ± 0.004	0.362	0.204
	FP	0.006 ± 0.003	0.573	0.032
UCM_‖_	BP	0.006 ± 0.011	0.222	0.445
	PP	0.005 ± 0.011	0.267	0.356
	FP	0.005 ± 0.012	0.301	0.296
UCM_Ratio_	BP	−0.032 ± 0.138	0.007	0.982
	PP	0.002 ± 0.141	0.173	0.555
	FP	−0.080 ± 0.124	0.139	0.635

## Data Availability

The data presented in this study are available in [20].

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
