# Peer review of "Running-Induced Fatigue Changes the Structure of Motor Variability in Novice Runners"

_biology, 2022, doi:10.3390/biology11060942_

Round 1

Reviewer 1 Report

As attached notes

Author Response

We thank the reviewer for the time and effort invested in revising our paper. Please find our responses in the attached Word-document.

Reviewer 2 Report

The paper applies the classical method of movement variability and movement control to novices runners. The paper is methodologically sound in the instruments (both hardware than modelization) proposed.

However some issue arises.

The abstract must be more informative and contain some data .

The COM is a theoretical conctruct, this must be stated. Also because as you know, it exist a whole biomechanical school of thinking which is against the use of COM.

In the abstract the authors states that is surprising that running COM studies consider mostly trained athletes. Is not clear why they should consider novice runners. There is no interest in doing that,  neither scientifically, neither practically. Studying novice runners is an ossimore.

Line 38. Runner are increasing in recent years ? The trends goes forward since 100 years at least.

line 44: explain why the studies are inconsistent.

line 50. Provide some further reference about the connection of injuries with the COM. 

line 67. Running speed means frequency or lenght ? Because above you wrote about adjuments of stride lenghts.

line 134. are these novice running 14 min / week or it is a typo error ? 

14 min per week is really less than novices.

Equation source. The proposed equation are based on some previously proposed equations  for COM calculation, or you wrote it newly ?

The same for the following equations, especially the UCM equations. IF exist please report the source or the basis on which you have worked on.

line 186. chosing the angle is good. But due that there are many other several contrains, why you don't chose a mix between angles, kinematics and so on ? why include only the angles ? please explain this point in the text.

line 225  the chosen levels of interval of CC is a general rules. However must be considered that we are in the specificity of human motion, so a certain differences in the proposed interval must be. Please look at some references about the definition of CC in human movement studies . 

line 233 better 6.18 without 0

Table 1. It is surprising only BP and FP are significant. Please give an hyphotesis for this.

Also line 269, BP moderatly correlated....this seems a weak point.

The limits of the Borg Scale must be evidenced in the methods. 

Overall the paper is interesting and acceptable for the publication after some revision and explanations

Author Response

(The authors gave the same response as above.)

Reviewer 3 Report

The manuscript was designed to analyze the influence of fatigue on stride-to-stride variability structure performed to stabilize the trajectory of the CoM in novice runners. 

Some points are difficult to understand in the reference. For instance, the authors claimed that Rabita et al. (2013) reported fatigue under fatigue at a constant speed. Then, they mention “in a similar study” in the following sentence but refer to the same one. Since the information derives from the same study, I am confused as they are opposite. 

It is also intriguing to follow the rationale applied by the authors. In the early stages, they mentioned: “The goal of runners, no matter if at novice, recreational or expert level, is to improve…” (Line 40). Then, in the next paragraph, they mentioned that “research concerning novice runners is sparse.” Please explain the rationale for studying novice runners since your arguments are conflicting. I am not sure if novice runners are more prone to injury as they have less exposure (mileage) than their more experienced peers. Furthermore, most arguments in the fourth paragraph fail your statements that expertise level is not relevant and that research with novice runners is sparse.

Motor variability or just variability?

It is also confusing the statements regarding the CoM. The authors mentioned that fatigue does not affect the spatio-temporal parameters or variability in novice runners, although the position of the CoM is affected (Lines 58-60). The following sentence states, "…and the associated CoM trajectory constant even when fatigued”. This isn't very clear. Revise.

Replace “body height” with “stature” and “body weight” with “body mass.”

Performing the study on a treadmill is dubious as the participants can not reduce speed. This is the opposite of what happens in a real-world scenario, as performers can adjust the pace according to their fatigue. Please comment.

Howe were the “last” cycles defined? Were they defined as before the end of the test? How was it determined? I understand you have used the Borg Scale, but it is unclear how it was applied to stop the test.

I have reservations regarding using correlations to relate variables, as they are not causal. In addition, oi your best scenario (r = 0.573), your correlations indicate that variance is explained by only ~33%. 

Some statements are tricky. For instance, “Increased variability in the orthogonal direction was compensated by the increased variability in the parallel direction, quantified with UCM‖, although this did not reach the level of statistical significance.” It is misleading to induce the reader to assume that an “increase” occurred while there was no statistical significance. No changes remove the perspective to name hope as “increases.” It is misleading.

I suggest authors plot the time normalized UCM‖ vs. UCMê“•, so the reader can understand how they interact. 

It is not clear how the authors determined the correlation of the Borg scale with the UCMê“•. Please, provide details.

The sentence “Although the PRE and POST states did not differ significantly, a tendency was observed showing that the curve was flattened.” Signals for another hope with no statistical strength to support it.

I wonder how the conclusion that CoM was controlled rather than joint configurations, based on the fact that the CoM relies on the joint's actions throughout the cycle. Please, explain.

The number of repetitions is critical depending on the instruction passed on to the performers. Did your participants receive any instructions regarding running and the task regarding what or how they were supposed to perform?

As a final caveat, I have to highlight that running at constant speed differs from what is seen in real work running (e.g., street running), where the participants can control and modify their paces regarding their fatigue level. It is only slightly commented and requires a more profound and thorough comment. 

What does the “stability” of the CoM refer to? Stability and variability seem to be used as the same terms. 

Author Response

(The authors gave the same response as above.)

Round 2

Reviewer 1 Report

In view of the adjustments presented, I consider the manuscript in a position to be published

Author Response

We thank the reviewer for this positive feedback.

Reviewer 3 Report

I thank the authors for their comprehensive responses. Although comprehensive, several aspects still need more detailed explanation or consideration.

First, exposure must be recognized as a factor as those with more mileage are likely to have more pronounced effects. I am not refuting the arguments (facts) reported by Kemler and collaborators. I suggest the authors acknowledge that wearing the structures is also a relevant component that can not be neglected. I also invite the authors to develop further the explanation provided by Kemler – if applicable. Several arguments provided in the response could be transposed to the manuscript to improve and strengthen the rationale.

Stature, please!

Yes, treadmills are fantastic but also misleading, irrespective if widely used or not. In a real-world scenario, no one keeps the speed constant when walking, running, or sprinting in an exhausted condition. Standardizing can be performed using any data, including non-representative ones. This is a difficult decision! I advocate in favor of real-world conditions rather than those “fabricated” in our labs. Indeed, runners may reduce speed to sustain their CoM characteristics. Is it true? I suggest adding this as a limitation. 

I insist that correlations are not causal. Please, reconsider specifically the amount of variance explained from the data, as pointed out in the previous round.

We mistakenly built this argumentation based on the “less changed” UCM_Ratio compared with the other two UCM components, although it was statistically not supported. Therefore, we deleted this part from our manuscript.

Please, I insist that the authors keep in mind that no significance means NO EFFECT. So, I invite them to revise the lines 313-316. Everything is correlated, and a p-value does not tell much of this long story. 

Author Response

Again, we thank the reviewer for the time invested in revising our paper.

Round 3

Reviewer 3 Report

I thank te authors for their direct responses and I have no further comments.